# SnO Nanosheet Transistor with Remarkably High Hole Effective Mobility and More than Six Orders of Magnitude On-Current/Off-Current

**DOI:** 10.3390/nano15090640

**Published:** 2025-04-23

**Authors:** Kuan-Chieh Chen, Jiancheng Wu, Pheiroijam Pooja, Albert Chin

**Affiliations:** Department of Electronics Engineering, National Yang Ming Chiao Tung University, Hsinchu 300, Taiwan; chieh.ee11@nycu.edu.tw (K.-C.C.); leo509078.ee11@nycu.edu.tw (J.W.); pheiroijampooja2022@nycu.edu.tw (P.P.)

**Keywords:** effective mobility, transistor, SnO, passivation, UV anneal

## Abstract

Using novel SiO_2_ surface passivation and ultraviolet (UV) light anneal, a 12 nm thick SnO p-type FET (pFET) shows hole effective mobilities (µ_eff_) of more than 100 cm^2^/V·s and 31.1 cm^2^/V·s at hole densities (Q_h_) of 1 × 10^11^ and 5 × 10^12^ cm^−2^, respectively. To further improve the on-current/off-current (I_ON_/I_OFF_), an ultra-thin 7 nm thick SnO nanosheet pFET shows a record-breaking I_ON_/I_OFF_ of 6.9 × 10^6^ and remarkable µ_eff_ values of ~70 cm^2^/V·s and 20.7 cm^2^/V·s at Q_h_ of 1 × 10^11^ cm^−2^ and 5 × 10^12^ cm^−2^, respectively. This is the first report of an oxide semiconductor transistor achieving a hole effective mobility µ_eff_ that reaches 20% of that in single-crystal Si pFETs at an ultra-thin body thickness of 7 nm. In sharp contrast, the control SnO nanosheet pFET without surface passivation or UV anneal exhibits a small I_ON_/I_OFF_ of 1.8 × 10^4^ and a µ_eff_ of only 6.1 cm^2^/V·s at 5 × 10^12^ cm^−2^ Q_h_. The enhanced SnO pFET performance is attributed to reduced defects and improved quality in the SnO channel, as confirmed by decreased charges related to sub-threshold swing (SS) and threshold voltage (Vth) shift. Such a large improvement is further supported by the increased Sn^2+^ after passivation and UV anneal, as evidenced by X-ray photoelectron spectroscopy (XPS) analysis. The I_ON_/I_OFF_ ratio exceeding six orders of magnitude, remarkably high hole µ_eff_, and excellent two-month stability demonstrate that this pFET is a strong candidate for integration with SnON nFETs in next-generation ultra-high-definition displays and monolithic three-dimensional integrated circuits (3D ICs).

## 1. Introduction

Oxide semiconductors possess unique carrier transport characteristics [1]. The SnO_2_ n-type field-effect transistor (nFET) has achieved a superb high field-effect mobility (µ_FE_) [2] close to the single-crystal Si nanosheet nFET and phonon-scattering-limited two-dimensional (2D) MoS_2_ nFETs [3,4]. However, p-type FETs (pFETs) are indispensable in forming low-DC-power Complementary Metal–Oxide–Semiconductor (CMOS) logic integrated circuits (ICs). The development of p-channel oxide TFTs remains challenging, as hole transport is highly sensitive to film quality and can be easily degraded by structural defects [5]. Materials such as tin monoxide (SnO), copper oxide (Cu_2_O), and nickel oxide (NiO) have emerged as promising p-type channel candidates due to their intrinsic hole conduction properties and compatibility with low-temperature processing. Recent advancements have focused on enhancing hole mobility through interface engineering, bilayer structures, and the incorporation of high-κ dielectrics. In particular, SnO-based pFETs have demonstrated notable improvements in hole mobility and thermal stability, enabled by advanced deposition techniques and optimized interface quality. For example, low-temperature processing has facilitated the integration of SnO pFETs into flexible electronic platforms [6]. Che et al. obtained a µ_FE_ of 6.13–7.24 cm^2^/V·s without using high-temperature post-anneal and improved sub-threshold swing characteristics by reducing surface defect states using backchannel passivation with an Al_2_O_3_ film [7]. Cu_2_O has also attracted interest for its inherent p-type conductivity and suitability for transparent electronics. Innovations in deposition techniques, such as aerosol-assisted chemical vapor deposition (AACVD), have enabled the fabrication of high-quality Cu_2_O thin films [8]. NiO, known for its wide bandgap and chemical stability, has been explored as another viable p-type semiconductor. Liu et al. reported a remarkable 60-fold enhancement in hole mobility from 0.07 to 4.4 cm^2^/V·s, primarily attributed to the high areal capacitance of the Al_2_O_3_ dielectric and the high-quality NiO_x_/Al_2_O_3_ interface, compared to conventional SiO_2_-based dielectrics [9]. Among the various oxide semiconductors, SnO has garnered significant attention due to its small hole effective mass [10,11,12,13,14,15,16], which originates from the strong orbital overlap between Sn 5s and O 2p states near the valence band edge [13,14]. Nevertheless, the development of high-performance SnO pFETs is hindered by oxide defects and the co-existence of n-type SnO_2−x_ [13]. In addition, SnO faces practical challenges such as sensitivity to moisture, stoichiometry control, and unavoidable oxygen vacancies, which may form nonstoichiometric phases such as Sn_2_O_3_ or Sn_3_O_4_ [15].

In this work, the record-breaking performance of an oxide-semiconductor pFET is reported, where the SnO pFET achieves a record-breaking on-current/off-current (I_ON_/I_OFF_) of 6.9 × 10^6^. Such a high I_ON_/I_OFF_ is mandatory for low-DC-power ICs. Furthermore, the device exhibits remarkably high effective mobilities (µ_eff_) of ~70 cm^2^/V·s and 20.7 cm^2^/V·s at hole densities (Q_h_) of 1 × 10^11^ cm^−2^ and 5 × 10^12^ cm^−2^, respectively. This record high performance among oxide semiconductor pFETs is attributed to the successful passivation of SnO surface defects and ultraviolet (UV) light anneal [16,17]. The passivation layer not only protects the SnO channel from moisture degradation, but also minimizes defect states, resulting in improved carrier transport. This excellent SnO pFET can be further integrated with the record-high-mobility SnON nFET [2] on the back end of ICs, forming low-DC-power CMOS logic ICs and being useful for monolithic three-dimensional (M3D) ICs.

## 2. Materials and Methods

Four-inch p+ Si substrates were used. Then, 50 nm thick TiN was deposited and used as the gate electrode. The TiN was sputtered using a DC plasma source of 800 W, with an Ar flow rate of 100 sccm and a N_2_ flow rate of 5 sccm. Electron beam evaporation was employed to deposit a 45 nm thick HfO_2_ film and 6 nm thick SiO_2_ as the gate dielectric, with the deposition rate controlled at 0.2 Å/s. After deposition, the device underwent thermal anneal in a N_2_ environment at 400 °C for 1 h. After anneal, 7 nm thick SnO was sputtered using a DC plasma source at 25 W, with an Ar flow rate of 24 sccm, an O_2_ flow rate of 8 sccm, a chamber pressure of 7.6 mTorr, and a deposition rate of 0.1 Å/s. The device was annealed using rapid thermal anneal (RTA) at 200 °C in a N_2_ environment for 30 s. Next, 30 nm thick Ni was deposited on the SnO and served as the source and drain electrodes. Additionally, passivation layers consisting of a 5 nm thick SiO_2_ film, a 10 nm thick HfO_2_ film, and a 3 nm thick SiO_2_ film were deposited. The device had a channel length of 50 μm and a channel width of 500 μm. Finally, UV anneal was performed with 254 nm at a 79 mW/cm^2^ power and 185 nm at an 11 mW/cm^2^ power for 10 min. First-principles quantum mechanical calculations were conducted to study the electronic properties of SnO using Quantum-espresso, with both the generalized gradient approximation (GGA) and local-density approximations plus the Hubbard potential U (LDA + U) method.

## 3. Results

SnO has a lead oxide structure and crystallizes in the tetragonal P4/nmm space group. The structure is 2D and consists of one SnO sheet oriented in the (0, 0, 1) direction. In the SnO unit cell, each Sn^2+^ is bonded in a four-coordinate geometry to four equivalent O^2−^ atoms. Each O^2−^ is bonded to four equivalent Sn^2+^ atoms to form a mixture of corner- and edge-sharing OSn_4_ tetrahedra. First-principles quantum mechanical calculations using Quantum-espresso are used to study the band structure of SnO (Figure 1) and calculate the effective mass of the hole (m_h_*). SnO bandgaps of 0.62 eV and 0.18 m_0_ and an effective mass of the hole (m_h_*) at **Γ** points are obtained, as shown in Table 1.

Figure 2a represents oxygen vacancy formation or the dangling bonds in the SnO structure during film processing caused by a low oxygen partial pressure while using a metallic Sn target. Moreover, insufficient oxygen during deposition or anneal can prevent the complete oxidation of Sn, leaving excess Sn atoms that may occupy interstitial sites, as shown in Figure 2b. Figure 2c,d show that with SiO_2_ passivation, the oxygen dangling bonds of SiO_2_ at the SnO/SiO_2_ interface could potentially interact with oxygen-deficient sites in SnO. In Figure 2d, a SiO_x_ transition layer forms at the interface between SiO_2_ and SnO. This SiO_x_ originates from the low-temperature-deposited SiO_2_ side, either due to oxygen diffusion or the partial reduction of SiO_2_ during processing. The interface contains various charges, as follows: oxide-trapped charges deep in SiO_2_, fixed-oxide charges near the interface, and interface-trapped charges at the SiO_x_/SnO boundary. These charges can affect device behavior by shifting the threshold voltage and degrading carrier mobility. Such dielectric and interface charges have been well studied in metal/SiO_2_/Si MOS capacitors, which cause Coulomb scattering and FET mobility degradation in 2D material FETs.

Figure 3a,b show a schematic structural diagram and a transmission electron microscope (TEM) cross-sectional image of the passivated SnO device, respectively. From the TEM, the thickness of SnO is 7 nm, the bottom interfacial SiO_2_ is 6 nm, and the top passivated SiO_2_/HfO_2_/SiO_2_ layers are 5, 10, and 3 nm, respectively. The TEM image in Figure 3b was taken after anneal. Interface roughness can increase hole scattering, especially at a high charge density or effective field [19]. Interface charges at both the top and bottom SnO interfaces result in mobility degradation and a poor sub-threshold performance. To lower interface charge scattering, a 6 nm thick SiO_2_ layer is deposited in between the channel SnO and the gate dielectric HfO_2_ to reduce remote phonon scattering from the high-κ gate dielectric. After the deposition of HfO_2_ and SiO_2_, the device undergoes thermal anneal in a N_2_ environment at 400 °C for 1 h before SnO channel deposition. Thermal anneal at 400 °C in N_2_ improves the electrical quality of the HfO_2_ and SiO_2_ dielectrics, reduces interface trap states, and enhances mobility. To further lower top interface charge scattering, SiO_2_ passivation is also applied.

Figure 4 shows the capacitance density versus voltage (C-V) and current density versus voltage (J-V) characteristics of the Ni/6 nm SiO_2_/45 nm HfO_2_/TaN metal–insulator–metal (MIM) structure, which was made using the same mask as the SnO pFET. The capacitance density reaches 260 nF/cm^2^, which gives an overall high dielectric constant (high-κ) value of 15.0. The gate leakage current is only 4 × 10^−6^ A/cm^2^, attributed to the high-κ gate dielectric. The gate leakage current of metal-gate/high-κ/SnO is significantly higher than that in standard metal-gate/high-κ/single-crystal Si Complementary Metal–Oxide–Semiconductor (CMOS) FETs [20]. This is because the former is made at a limited temperature of 400 °C for backend-of-line (BEOL) processes while the latter is fabricated at 1000 °C. The limited thermal budget causes defects and trap-assisted leakage in the high-κ gate dielectric. To further decrease gate leakage current under low-temperature processing conditions, atomic layer deposition (ALD) can be employed; however, this involves a relatively long process time.

The FET’s µ_FE_ and µ_eff_ are obtained under a small drain voltage (V_d_) [21], as follows:(1)$$\mu_{FE}=\frac{L}{W}\frac{1}{C_{OX}V_{d}}\frac{{dI}_{d}}{{dV}_{g}}{|}_{small\;V_{d}}=\frac{L}{W}\frac{1}{C_{OX}V_{d}}g_{m}{|}_{small\;V_{d\;}},$$(2)$$\mu_{eff}=\frac{L}{W}\frac{g_{d}}{qN_{inv}}{|}_{small\;V_{d}}=\frac{L}{W}\frac{g_{d}}{C_{OX}{(V}_{gs}-V_{th})},$$
where L and W are the channel length and width, C_ox_ is the gate capacitance per unit area, gm is the transconductance, q_Ninv_ is the total induced Q_h_ in the channel, and g_d_ is the drain conductance, respectively. Here, µ_eff_ is essential for transistor modeling and crucial for IC design.

Figure 5a,b present the drain current versus V_d_ (|I_d_| − V_d_) characteristics of nanosheet FETs, with 10 and 12 nm thick SnO channels, respectively. Both transistors show an increase in |I_d_| with an increasingly negative gate voltage (V_g_), which indicates the pFET’s behavior. A good transistor pinch off is observed at a positive V_g_. Additionally, the output |I_d_| is higher for the thicker SnO channel, suggesting enhanced hole conduction in the thicker SnO pFET. In Figure 5, the drain current continues to increase in the saturation regime. This increasing trend is not due to the gate leakage current, which is significantly lower than the drain current. Instead, it is attributed to the thicker SnO channel, which cannot be fully depleted by the gate electric field. To further reduce the gate leakage current under low-temperature processing conditions, ALD is a useful technique, although it involves a relatively long processing time.

Figure 6a–c show the |I_d_| − V_g_, µ_FE_ − V_g_, and µ_eff_ − Q_h_ characteristics for 10 and 12 nm thick SnO pFETs. The 12 nm thick SnO pFET shows a superb transistor hole µ_eff_ of more than 100 cm^2^/V·s and 31.1 cm^2^/V·s at a Q_h_ of 1 × 10^11^ and 5 × 10^12^ cm^−2^, respectively. At a high 5 × 10^12^ cm^−2^ Q_h_, it is crucial to notice that the hole µ_eff_ increases from 27.3 cm^2^/V·s for the 10 nm thick SnO to 31.1 cm^2^/V·s for the 12 nm channel thickness. Although a higher µ_eff_ and output |I_d_| are obtained with the thicker SnO pFETs, there is a significant degradation in the I_OFF_. The large I_OFF_ causes too high off-state leakage current and DC power consumption, which is intolerant for modern ICs with multi-billions of FETs. The high I_OFF_ is attributed to the inability of the gate and surface potentials to fully deplete the thicker SnO channel. It is also noted that UV anneal does not alter the SnO film thickness, as confirmed by similar experiments performed on both SnO_2_ nFETs [16] and SnO pFETs [17].

Figure 7a–c show the |I_d_| − V_d_ characteristics for thinner 7 nm thick SnO pFETs under different device conditions, with and without passivation and with and without UV anneal. The increase in I_d_| with more −V_g_ confirms the p-type behavior of the FETs. At a V_g_ of −2 V, the output |I_d_| increases from the control device to the device with surface passivation and the pFET with both passivation and UV anneal. The |I_d_| curves do not increase monotonically with V_G_, which is due to the roll off of µ_eff_ with an increasing Q_h_ [19].

Figure 8a–c present the |I_d_| − V_g_ characteristics of the 7 nm thick SnO device with and without passivation and with and without UV anneal. At V_d_ = −0.1 V, the SnO pFET with a passivation layer and UV anneal shows the highest |I_d_|. Such a higher |I_d_| is important for a higher-speed IC device.

Figure 9a shows the comparison of the |I_d_| − V_g_ characteristics for 7 nm thick SnO pFETs without UV anneal. The non-passivated control device exhibits an I_ON_/I_OFF_ of 1.8 × 10^4^, whereas the passivated device achieves a large 1.1 × 10^6^. The threshold voltages (Vth) extracted from the extrapolation of the I_d_ linear region [22] are −0.1 and 0.2 V for the FETs without and with passivation, respectively. The two-orders-of-magnitude I_ON_/I_OFF_ improvement suggests that the passivation effectively isolates the channel surface from reacting with moisture. The SnO surface reaction with moisture further forms charged defects, which change the Vth and decrease the I_ON_. Figure 9b shows the comparison of the |I_d_| − V_g_ characteristics between the control FET and the best-performing FET with passivation and UV anneal. The UV anneal further increases the I_ON_/I_OFF_ from 1.1 × 10^6^ to 6.9 × 10^6^. Such an improvement is attributed to the UV anneal effectively enhancing channel quality, thereby resulting in a superior electrical performance [17].

Figure 10a,b illustrate the µ_FE_ − V_g_ and µ_eff_ − Q_h_ characteristics for 7 nm thick SnO pFETs, respectively. The peak µ_FE_ values increase from 5.8 and 14.3 to 20.3 cm^2^/V·s. The µ_eff_ values decrease from ~70 m^2^/V·s with an increasing Q_h_. At a Q_h_ of 5 × 10^12^ cm^−2^, µ_eff_ increases from 6.1 and 14.7 to 20.7 cm^2^/V·s. This is the first time that the hole µ_eff_ of an oxide semiconductor has reached 20% of that of a single-crystal Si pFET at a 7 nm ultra-thin body thickness [23]. This improved pFET performance is due to both passivation and UV anneal, resulting in reduced defects and an improved quality of the SnO channel. Moreover, the interfacial charge trap density (Dit) [24] is calculated, and we find that passivated SnO with UV anneal has a Dit of 4.6 × 10^12^ cm^−2^ eV^−1^, close to that of passivated SnO (4.7 × 10^12^ cm^−2^ eV^−1^) and better than the SnO (8.2 × 10^12^ cm^−2^ eV^−1^). As shown in Figure 10a, the shoulder-like non-smooth µ_FE_ feature in the 7 nm thick SnO channel is attributed to extra scattering from defects, which can be improved by passivation and UV anneal. The shoulder-like non-smooth µ_FE_ feature in Figure 6a,b is due to a thicker SnO channel, where gate electric field cannot fully deplete the SnO channel and defects. This is confirmed by the higher I_ON_/I_OFF_ in thicker SnO channel pFETs.

Figure 11 presents the |I_d_| − V_g_ characteristics of 7 nm thick SnO pFETs as-fabricated and after two months of exposure to air. The control device shows significant deterioration after exposure to air, and the µ_FE_ decreases from 5.8 to 1.5 cm^2^/V·s after 2 months. In contrast, the device with passivation exhibits minimal changes in electrical performance as follows: the passivated and non-UV annealed device decreases from 14.3 to 12.3 cm^2^/V·s, and the passivated and UV-annealed device decreases slightly from 20.3 to 19.2 cm^2^/V·s. This comparison highlights the significant impact of passivation and channel material quality on device performance. The device shown in Figure 11a is not encapsulated to show the aging effect. The unpassivated control device exhibits significant degradation after two months of air exposure, with the μ_FE_ dropping from 5.8 to 1.5 cm^2^/V·s. In contrast, the passivated device only shows a minimal decline from 20.3 to 19.2 cm^2^/V·s, highlighting the strong protective role of a passivation approach. For future practical integration into ICs, external encapsulation using SiN will be applied to top Cu and low-κ interconnecting BEOL layers to improve long-term stability. However, SiO_2_ passivation directly on SnO pFETs must still be applied to prevent any moisture-related degradation during BEOL processes.

To further understand such an excellent performance, the SnO channels are characterized by X-ray photoelectron spectroscopy (XPS) [25,26]. The XPS results are calibrated using the C1s binding energy peak at 284.4 eV as a reference. Figure 12 shows the XPS Sn 3d_5/2_ peaks for SnO pFETs under different conditions. The Sn 3d_5/2_ peaks can be categorized into the following three types: Sn^0^ at 484.1 eV, Sn^2+^ at 486.3 eV, and Sn^4+^ at 486.9 eV [16,27]. Table 2 summarizes the deconvoluted Sn 3d_5/2_ XPS of Sn^0^, Sn^2+^, and Sn^4+^. Because the lattice energies of SnO_2_ (11,807 kJ mol^−1^) are much higher than those of SnO (3652 kJ mol^−1^), surface SnO can react with oxygen and moisture in air to form SnO_1+x_ [28]. Consequently, when passivation and UV anneal are applied, the proportion of Sn^2+^ increases from 65.2% to 73.8% and Sn^4+^ decreases from 27.7% to 16.5%.

Table 3 summarizes the performance comparison between this study and other SnO pFET devices published in the literature. By applying passivation and UV anneal to SnO, the sub-threshold swing (SS) value is improved to 231 mV/dec compared with SnO (358 mV/dec) and passivated SnO (233 mV/dec). Our SnO pFETs provide the best I_ON_/I_OFF_ near seven orders of magnitude, one of the lowest I_OFF_, a small SS, and a remarkably high µ_eff_ for high-speed and for low-power applications. Device reproducibility has been demonstrated in our previous publications [13,17,29]. The μ_eff_ of 20.7 cm^2^/V·s is the highest value obtained for a SnO pFET with a typical standard deviation of 3.4 for 10 devices [13] and an I_ON_/I_OFF_ of 6.9 × 10^6^ with a standard deviation of 87. This level of device variability is comparable to that observed in our previously reported SnON nFETs [2].

## 4. Conclusions

In this study, we investigated the electrical performance and material properties of the following three types of devices: devices without a passivation layer and without UV anneal, devices with a passivation layer but without UV anneal, and devices with both a passivation layer and UV anneal. From XPS, the SnO with a passivation layer and UV anneal showed a reduced Sn^4+^ ratio and increased Sn^2+^, which resulted in an I_ON_/I_OFF_ as high as 6.94 × 10^6^ and an effective mobility of up to 20.7 cm^2^/V·s at a hole density (Q_h_) of 5 × 10^12^ cm^−2^ in the pFET. These results indicate that this nanosheet SnO FET meets the requirements of simple, low-cost, and low-temperature fabrication, and will be useful for future monolithic 3D CFET applications.

## Figures and Tables

**Figure 1 nanomaterials-15-00640-f001:**
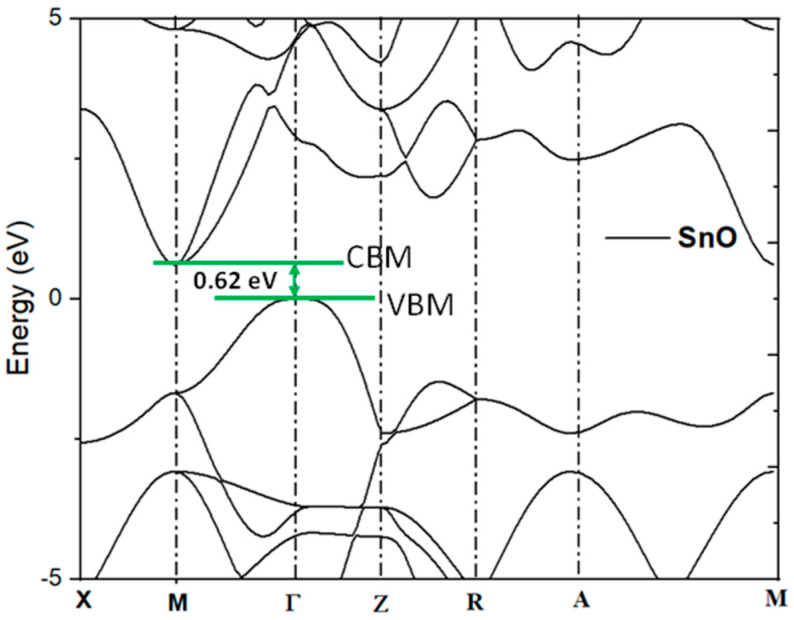
Energy band structure of SnO. CBM and VBM represent conduction band minimum and valence band maximum, respectively.

**Figure 2 nanomaterials-15-00640-f002:**
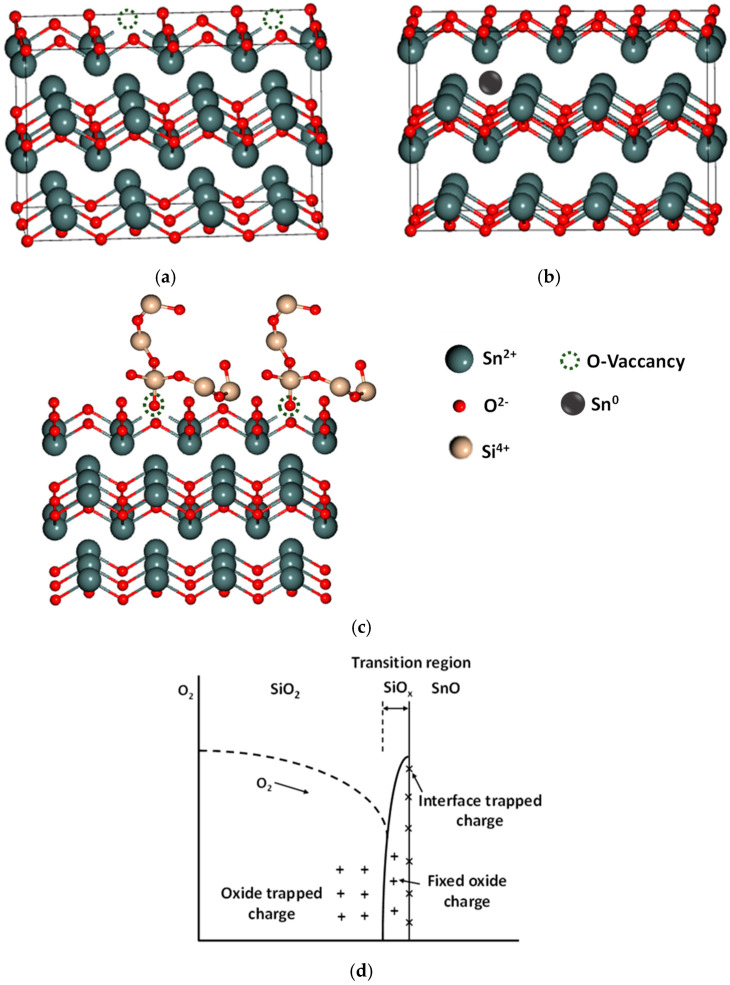
SnO structure with (**a**) oxygen vacancies (dangling bonds), (**b**) Sn interstitial, (**c**) SiO_2_ passivation, and (**d**) interface charges.

**Figure 3 nanomaterials-15-00640-f003:**
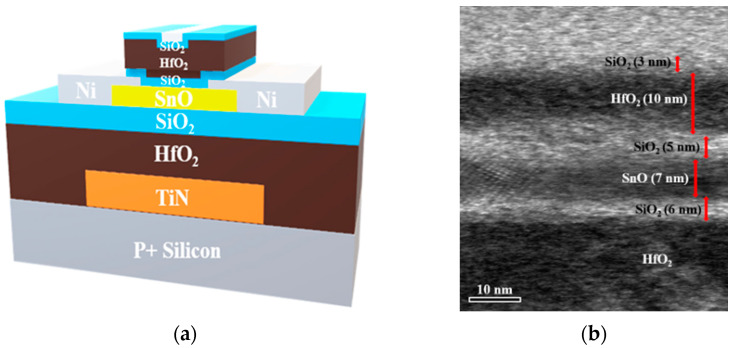
(**a**) Schematic diagram and (**b**) TEM cross-sectional image of the SnO transistor structure with a passivation layer.

**Figure 4 nanomaterials-15-00640-f004:**
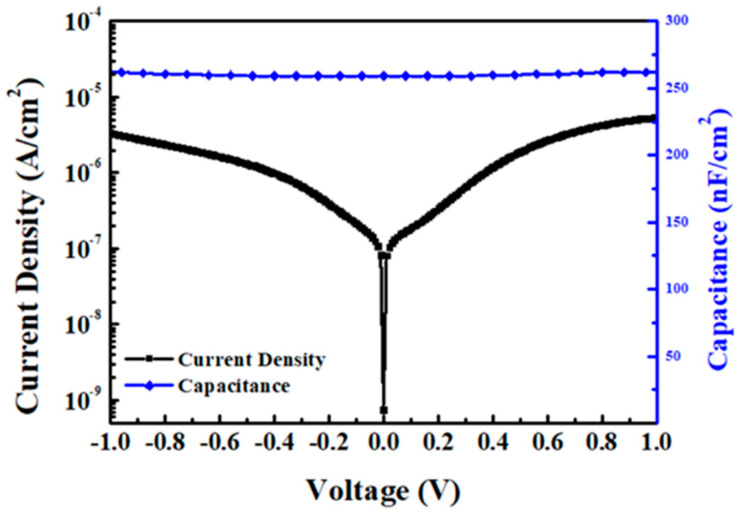
C-V at 1 kHz and J-V characteristics of the Ni/6 nm SiO_2_/45 nm HfO_2_/TaN capacitor.

**Figure 5 nanomaterials-15-00640-f005:**
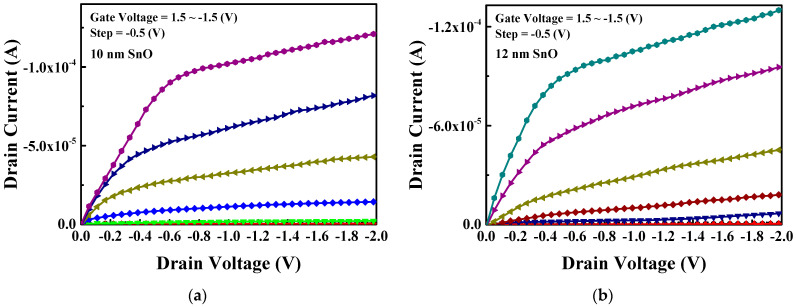
|I_d_| − V_d_ characteristics of passivated and UV-annealed SnO nanosheet pFETs with (**a**) 10 nm and (**b**) 12 nm thick SnO channel.

**Figure 6 nanomaterials-15-00640-f006:**
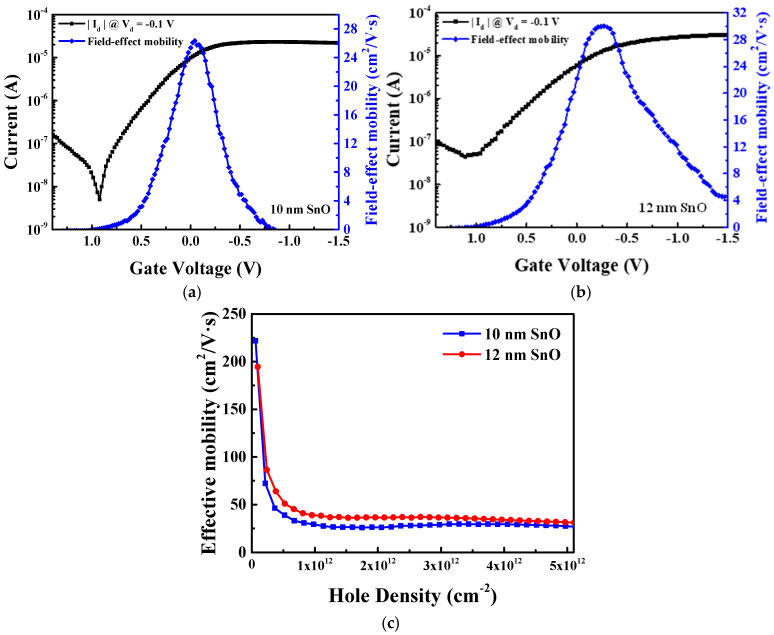
(**a**) |I_d_| − V_g_ and µ_FE_ − V_g_ characteristics of (**a**) 10 nm and (**b**) 12 nm thick SnO channel FETs. (**c**) µ_eff_ − Q_h_ characteristics of passivated and UV-annealed pFETs with 10 and 12 nm SnO thicknesses.

**Figure 7 nanomaterials-15-00640-f007:**
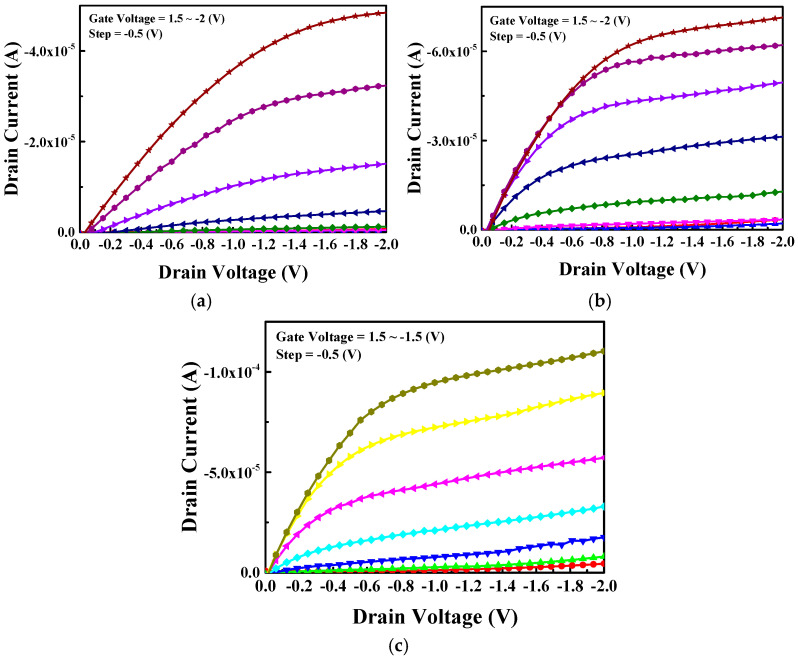
|I_d_| − V_d_ characteristics for 7 nm thick SnO pFETs (**a**) without passivation and UV anneal, (**b**) with passivation but without UV anneal, and (**c**) with passivation and UV anneal.

**Figure 8 nanomaterials-15-00640-f008:**
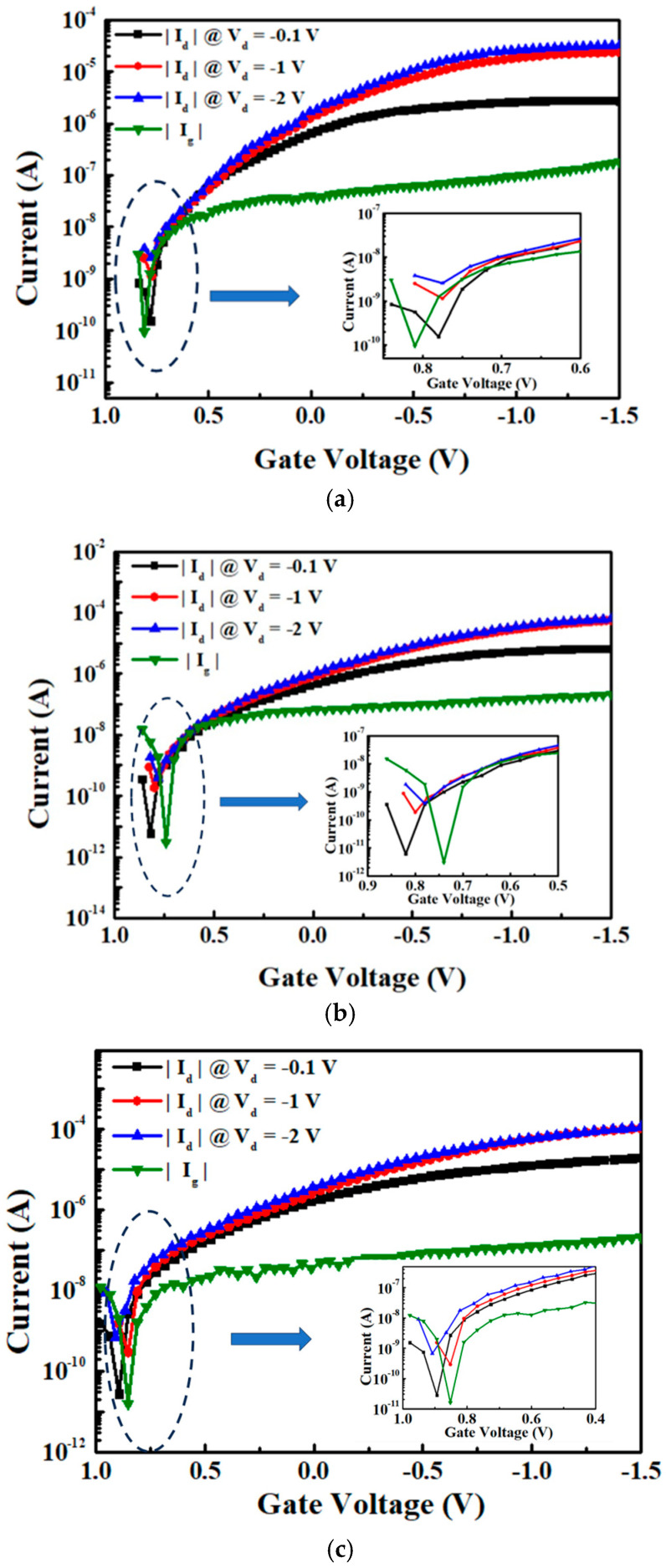
|I_d_| − V_g_ characteristics for 7 nm thick SnO pFETs in linear and saturation region of (**a**) control device without passivation or UV anneal, (**b**) with passivation but without UV anneal, and (**c**) with passivation and UV anneal (inset shows magnified |I_d_| and |I_g_| near I_OFF_).

**Figure 9 nanomaterials-15-00640-f009:**
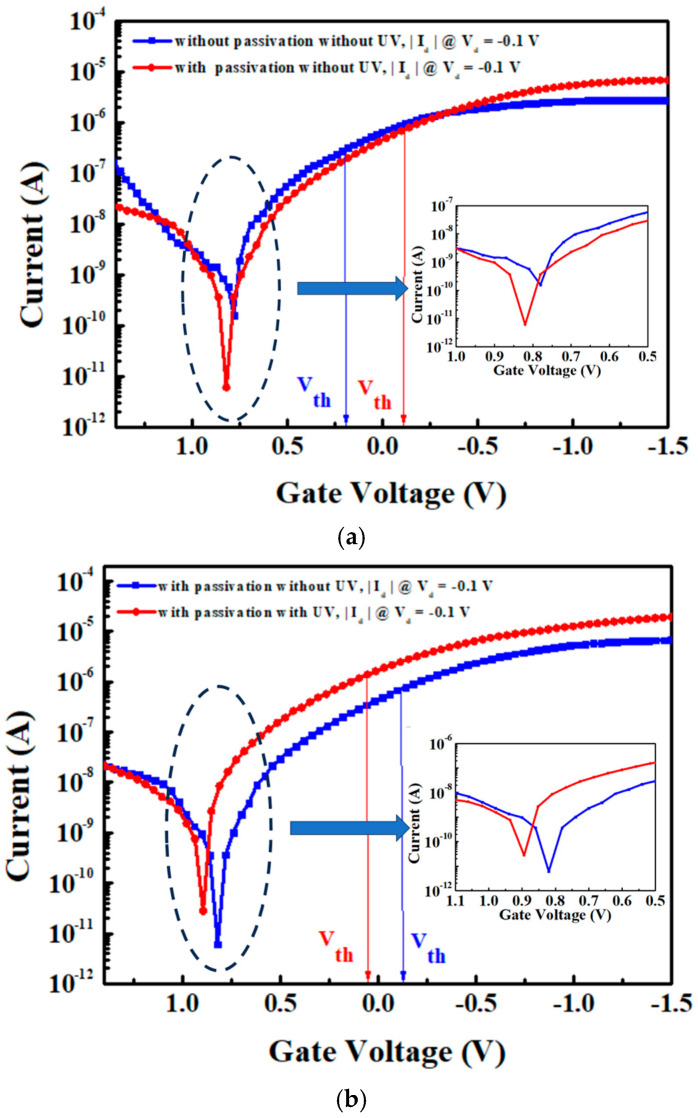
Comparison of |I_d_| − V_g_ characteristics for 7 nm thick SnO pFETs (**a**) without (control) and with passivated pFETs, and (**b**) control and the best device with passivation and UV anneal (inset shows magnified |I_d_| near I_OFF_).

**Figure 10 nanomaterials-15-00640-f010:**
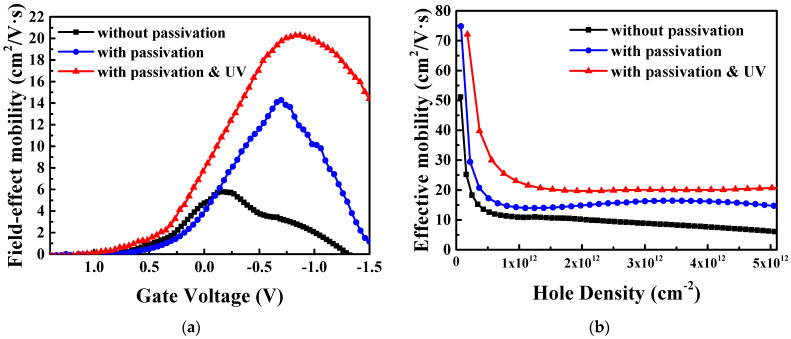
(**a**) µ_FE_ − V_g_ and (**b**) µ_eff_ − Q_h_ characteristics for 7 nm thick SnO pFETs.

**Figure 11 nanomaterials-15-00640-f011:**
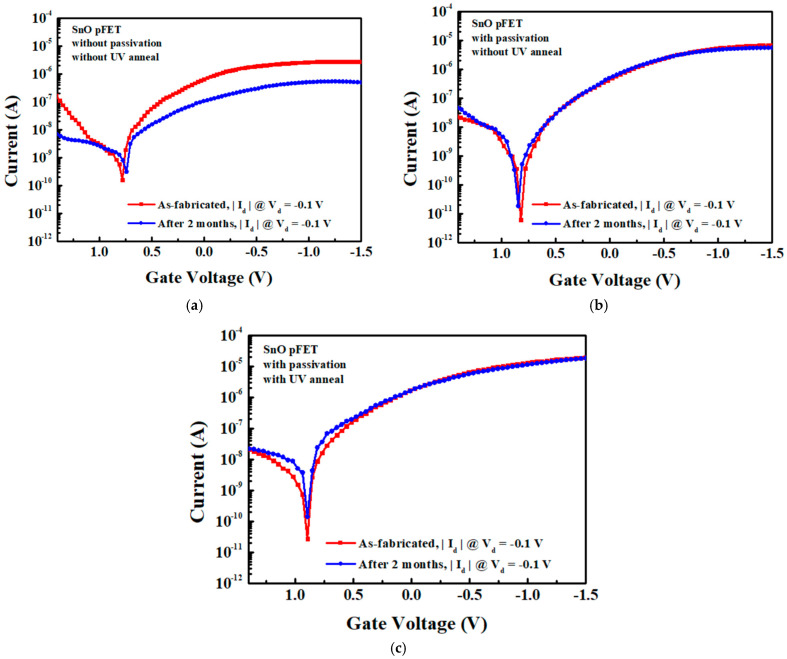
|I_d_| − V_g_ characteristics for 7-nm thick SnO pFETs (**a**) control device without passivation or UV anneal, (**b**) with passivation but without UV anneal, and (**c**) with passivation and with UV anneal. The devices were measured as-fabricated and after exposure to air for two months.

**Figure 12 nanomaterials-15-00640-f012:**
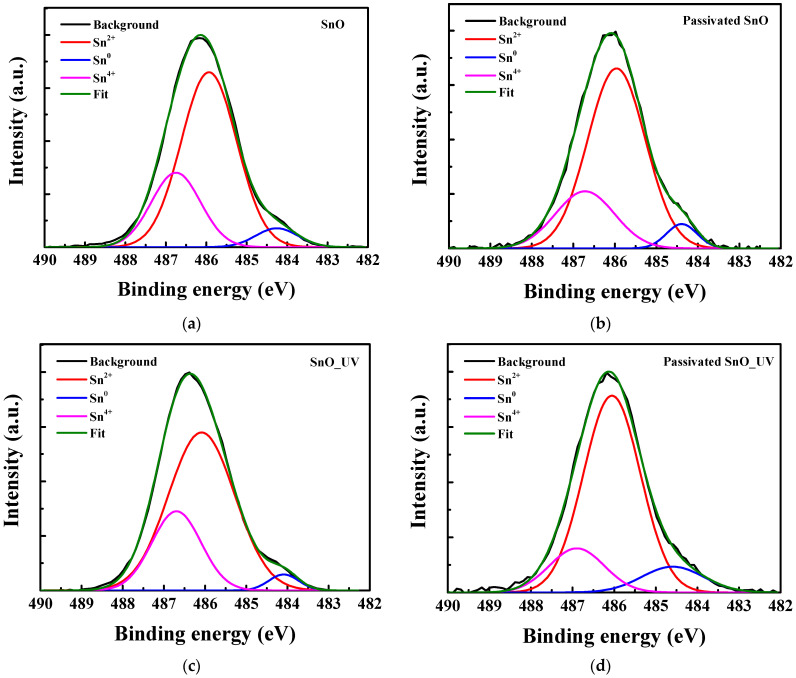
XPS Sn 3d_5/2_ peaks of SnO pFET (**a**) w/o passivation w/o UV, (**b**) w/o passivation w/o UV, (**c**) w/i passivation w/o UV, and (**d**) w/i passivation w/i UV.

**Table 1 nanomaterials-15-00640-t001:** Comparison table of p-type metal–oxide–semiconductors.

P-Type	E_g_	Hole Effective Mass (m_h_*)
SnO	0.62 eV (Indirect)	0.18 m_0_
Cu_2_O [13]	2.1 eV (Direct)	0.65 m_0_
NiO [18]	3.6 eV (Indirect)	0.45 m_0_

**Table 2 nanomaterials-15-00640-t002:** Summary of Sn3d_5/2_ proportions.

Sample	Sn^0^	Sn^2+^	Sn^4+^
SnO	7.1%	65.2%	27.7%
SnO_UV	6.3%	62.4%	31.3%
Passivated SnO	9.3%	68.9%	21.8%
Passivated SnO_UV	9.7%	73.8%	16.5%

**Table 3 nanomaterials-15-00640-t003:** Comparison of SnO pFET performance.

Refs.	Channel	I_OFF_ (A/μm)	I_ON_/I_OFF_	SS (mV/dec)	µ_FE_ (cm^2^/V·s)	µ_eff_ (cm^2^/V·s)	Tech. and Anneal
This work	SnO	5.6 × 10^−14^	6.9 × 10^6^	231	20.3	20.7	200 °C RTA SiO_2_ passivation UV anneal
[12]	SnO	-	~10^2^	-	1.3	-	200 °C RTA
[10]	SnO	~10^−12^	6 × 10^3^	7630	6.75	-	180 °C 30 min
[11]	SnO	~4 × 10^−10^	>10^3^	760	10.83	-	160 °C
[30]	SnO	-	2.7 × 10^2^	4600	6	-	300 °C 1 h
[31]	SnO	-	6.54 × 10^5^	150	1.14	-	300 °C 1 h
[32]	SnO	-	2.5 × 10^3^	240.9	38.7	-	175 °C 2 h
[17]	SnO	~2.5 × 10^−13^	1.05 × 10^5^	274	13.33	13.38	200 °C RTA UV anneal
[33]	Cu_2_O	-	4.68 × 10^4^	3910	1.11	-	Thermal anneal 800 °C 1 h
[5]	Cu_2_O	-	4.1 × 10^6^	2350	1.38	-	100 °C 1 h
[34]	NiO	2.75 × 10^−7^	3.61 × 10^4^	-	1.09	-	200 °C 30 min
[9]	NiO	-	10^5^	250	4.4	-	250 °C
[35]	SnO	-	4.2 × 10^3^	6100	17.2	19.1	300 °C 1 h
[36]	NiO	-	10^3^	2600	1.07	-	-
[37]	NiO_x_	-	10^5^	700	25	-	UV treatment, 40 min and anneal 350 °C 1 h
[38]	Cu_2_O	-	3.4 × 10^2^	26,000	0.9	-	700 °C
[39]	Cu_2_O	-	-	-	4.3 × 10^−2^	-	650 °C oxidation
[40]	Cu_2_O	-	10^2^	-	0.16	-	400 °C 30 min
[41]	Ga-doped Cu_2_O	-	1.22 × 10^4^	7720	0.74	-	800 °C

## Data Availability

The data presented in this study are available on request from the corresponding author. The data are not publicly available due to privacy.

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
