# Peer review of "SnO Nanosheet Transistor with Remarkably High Hole Effective Mobility and More than Six Orders of Magnitude On-Current/Off-Current"

_nanomaterials, 2025, doi:10.3390/nano15090640_

Round 1

Reviewer 1 Report

Comments and Suggestions for Authors

The manuscript titled "SnO Nanosheet Transistor with Remarkably High Hole Effective Mobility and More Than Six Orders of Magnitude On-current/Off-current" written by Kuan-Chieh Chen et al. reported for the first time a 7 nm SnO nanosheet pFET synergistically optimized with SiO₂ passivation and UV annealing, achieving a record-breaking on-current/off-current ratio (ION/IOFF) of 6.9×10⁶ and an effective hole mobility (μeff) of 20.7 cm² V⁻¹ s⁻¹. Remarkably, the μeff reaches 20% of that in single-crystal Si pFETs, marking a milestone in ultra-thin-body oxide pFETs. While the results demonstrate significant potential and reference value, several issues must be addressed to strengthen the manuscript before publication.

I recommend that this manuscript be published in Nanomaterials if the following issues are addressed:

  1. Performance Comparison Table: A table comparing key performance metrics (μeff, ION/IOFF, threshold voltage shift (Vth), and subthreshold swing (SS)) with existing oxide pFETs (e.g., Cu₂O, NiO) should be included to contextualize the novelty of this work.
  2. Verification of "First-Time Achievement": A comprehensive literature review is required to confirm the absence of prior reports on SnO pFETs, particularly those published after 2022. If overlapping work exists, the novelty statement should be revised with appropriate citations.
  3. Repeatability and Statistical Significance: Statistical results for μeff and ION/IOFF (including standard deviation and confidence intervals) from at least three devices should be provided to demonstrate process stability.
  4. Control Group Experiment: The SnO thickness of the control group (without passivation/UV annealing) must be clarified to ensure that thickness variations do not influence performance comparisons.
  5. Figure 1 Band Structure: The position of the Γ point in the Brillouin zone, as well as the valence band maximum and conduction band minimum, should be explicitly labeled to align with the theoretically calculated bandgap of 0.62 eV.
  6. Figures 7–9 Current Curves: Logarithmic-scale Id-Vg curves (e.g., as insets) should be added to clearly illustrate the span of ION/IOFF.
  7. Figure Organization: The figures should be reorganized for better visual flow and clarity.
  8. Terminology Consistency: Expressions such as "UV anneal" and "UV annealing" should be standardized throughout the manuscript.
  9. Reference Updates: The introduction should include recent advancements in oxide pFETs from the past five years to ensure the discussion reflects the latest developments.
Comments on the Quality of English Language

The English expression of this manuscript flows smoothly, effectively conveys its ideas without causing ambiguity, and stands as an excellent academic paper.

Author Response

Thank you for the comments. To strengthen this paper, we have included your important comments in the revised manuscript (highlighted with yellow color).

  1. Performance Comparison Table: A table comparing key performance metrics (μeff, ION/IOFF, threshold voltage shift (Vth), and subthreshold swing (SS)) with existing oxide pFETs (e.g., Cu₂O, NiO) should be included to contextualize the novelty of this work.

Ans:   As suggested, the following comparison table including Cu2O and NiO has been added in the revised manuscript.

Refs.

Channel

IOFF

(A/μm)

ION/IOFF

SS (mV/dec)

µFE

(cm2/V·s)

µeff

(cm2/V·s)

Tech. & Anneal

This work

SnO

5.6×10-14

6.9×106

231

20.3

20.7

200°C RTA

SiO2 passivation

UV anneal

[13]

SnO

-

~ 102

-

1.3

-

200°C RTA

[11]

SnO

~ 10-12

6×103

7630

6.75

-

180°C 30min

[12]

SnO

~ 4×10-10

> 103

760

10.83

-

160°C

[19]

SnO

~ 2×10-13

3×104

140

7.6

-

200°C

[35]

SnO

-

2.7×102

4600

6

-

300°C 1hr

[36]

SnO

~ 1.3×10-13

2×105

526

4.4

-

200°C 45min

[37]

SnO

-

6.54×105

150

1.14

-

300°C 1hr

[38]

SnO

-

2.5×103

240.9

38.7

-

175°C 2hr

[18]

SnO

~ 2.5×10-13

1.05×105

274

13.33

13.38

200°C RTA

UV anneal

[39]

Cu2O

-

4.68×104

3910

1.11

-

Thermal anneal 800°C 1hr

[6]

Cu2O

-

4.1×106

2350

1.38

-

100°C 1hr

[40]

NiO

2.75 ×10-7

3.61×104

-

1.09

-

200°C 30 min

[10]

NiO

-

105

250

4.4

-

250°C

[41]

SnO

-

4.2×103

6100

17.2

19.1

300°C 1hr

[6] Chang, H., Huang, C.H., Matsuzaki, K., Nomura, K., Back-channel defect termination by sulfur for p-channel Cu2O thin-film transistors, ACS Appl. Mater. Interfaces. 2020, 12(46), 51581-51588.

[10] Liu, A., Liu, G., Zhu, H., Shin, B., Fortunato, E., Martins, R., Shan, F., Hole mobility modulation of solution-processed nickel oxide thin-film transistor based on high-k dielectric, Appl. Phys. Lett. 2016, 108(23), 233506.

[39] Lee, J.H., Kim, J., Jin, M., Na, H.J., Lee, H., Im, C., Kim, Y.S., Cu2O p-type thin-film transistors with enhanced switching characteristics for CMOS logic circuit by controlling deposition condition and annealing in the N2 atmosphere, ACS Appl. Electron. Mater. 2023, 5(2), 1123-1130.

[40] Lin, C.W., Chung, W.C., Zhang, Z.D., Hsu, M.C., P-channel transparent thin-film transistor using physical-vapor-deposited NiO layer, Jpn. J. Appl. Phys. 2017, 57(1S), 01AE01.

[41] Mashooq, K., Jo, J., Peterson, R.L., Effect of metal capping layer in achieving record high p-type SnO thin film transistor mobility, IEEE Trans. Electron Devices. 2023, 71(1), 574-580.

  1. Verification of "First-Time Achievement": A comprehensive literature review is required to confirm the absence of prior reports on SnO pFETs, particularly those published after 2022. If overlapping work exists, the novelty statement should be revised with appropriate citations.

Ans: As suggested, after comprehensive review, we found that SnO pFETs is the transistor with hole µeff that reaches 20% of single-crystal Si pFET [22].

[22] Takagi, S.I., Toriumi, A., Iwase, M, Tango, H., On the universality of inversion layer mobility in Si MOSFET's: Part I-effects of substrate impurity concentration, IEEE Trans. Electron Devices. 1994, 41, 2357 – 2362.

  1. Repeatability and Statistical Significance: Statistical results for μeff and ION/IOFF (including standard deviation and confidence intervals) from at least three devices should be provided to demonstrate process stability.

Ans: The device reproducibility can be found in our previous publications [34], [14], [18]. The μeff of 20.7 cm2/V·s is the highest value for SnO pFET with a typical standard deviation of 3.4 for 10 devices and ION/IOFF of 6.9 × 10⁶ with standard deviation of 87. Such device variability was also close to our reported works on SnON nFET [2].

[2] Pooja, P., Che, C. C., Zeng, S. H., Lee, Y. C., Yen, T. J., Chin, A., Outstanding high field‐effect mobility of 299 cm2 V−1s−1 by nitrogen‐doped SnO2 nanosheet thin‐film transistor, Adv. Mater. Technol. 2023, 8(7), 2201521.

[14] Shih, C.W., Chin, A., Lu, C.F., Su, W.F., Remarkably high hole mobility metal-oxide thin-film transistors, Sci. Rep. 2018, 8(1), 889.

[18] Zeng, S.H., Pooja, P., Wu, J., Chin, A., Impact of UV annealing on the hole effective mobility in SnO pFET, Sci. Rep. 2024, 14(1), 26256.

[34] Yen, T.J., Chin, A., Gritsenko, Exceedingly high performance top-gate p-type SnO thin film transistor with a nanometer scale channel layer, Nanomaterials. 2021, 11, 92.

  1. Control Group Experiment: The SnO thickness of the control group (without passivation/UV annealing) must be clarified to ensure that thickness variations do not influence performance comparisons.

Ans: The UV anneal will not change the thickness as we have carried out similar experiments with UV anneal on SnO2 nFET [17] and SnO pFET [18] respectively.

[17] Shih, C.W., Chin, A., Lu, C.F., Su, W.F., Low-temperature processed tin oxide transistor with ultraviolet irradiation, IEEE Electron Device Lett. 2019, 40(6) 909-912.

[18] Zeng, S.H., Pooja, P., Wu, J., Chin, A., Impact of UV annealing on the hole effective mobility in SnO pFET, Sci. Rep. 2024, 14(1), 26256.

  1. Figure 1 Band Structure: The position of the Γ point in the Brillouin zone, as well as the valence band maximum and conduction band minimum, should be explicitly labeled to align with the theoretically calculated bandgap of 0.62 eV.

Ans: As suggested, the following changes has been updated in the revised manuscript.

Figure 1. Energy band structures of SnO. CBM and VBM represent conduction band minimum and valence band maximum respectively.

  1. Figures 7–9 Current Curves: Logarithmic-scale Id-Vg curves (e.g., as insets) should be added to clearly illustrate the span of ION/IOFF.

Ans: The logarithmic scale Id-Vg curves as inset has been added in the revised manuscript.

(a)

(b)

(c)

Figure 8. |Id|-Vg characteristics for 7-nm thick SnO pFETs in linear and saturation region of (a) control device without passivation nor UV anneal, (b) with passivation but without UV anneal, and (c) with passivation and UV anneal. (inset shows magnified |Id| and |Ig| near IOFF).

(a)

(b)

Figure 9. Comparison of |Id|-Vg characteristics for 7-nm thick SnO pFETs (a) without (control) and with passivated pFETs, and (b) control and the best device with passivation and UV anneal. (inset shows magnified |Id| near IOFF).

  1. Figure Organization: The figures should be reorganized for better visual flow and clarity.

Ans: As suggested, the figures have been checked in the revised manuscript.

  1. Terminology Consistency: Expressions such as "UV anneal" and "UV annealing" should be standardized throughout the manuscript.

Ans: As suggested, terminology consistency has been checked in the revised manuscript.

  1. Reference Updates: The introduction should include recent advancements in oxide pFETs from the past five years to ensure the discussion reflects the latest developments

Ans: Thank you for your suggestion. The following recent advancements in oxide pFETs has been included in the introduction section in the revised manuscript.

The development of p-channel oxide TFTs remains challenging, as hole transport is highly sensitive to film quality and can be easily degraded by structural defects [6]. Materials such as tin monoxide (SnO), copper oxide (Cu₂O), and nickel oxide (NiO) have emerged as promising p-type channel candidates due to their intrinsic hole conduction properties and compatibility with low-temperature processing. Recent advancements have focused on enhancing hole mobility through interface engineering, bilayer structures, and the incorporation of high-κ dielectrics. In particular, SnO-based pFETs have demonstrated notable improvements in hole mobility and thermal stability, enabled by advanced deposition techniques and optimized interface quality. For example, low-temperature processing has facilitated the integration of SnO pFETs into flexible electronic platforms [7]. Che et al. obtained field effect mobility of 6.13-7.24 cm2/V·s without using a high-temperature post-anneal and improved the subthreshold swing characteristics by reducing surface defect states using backchannel passivation by the Al2O3 film [8]. Cu₂O has also attracted interest for its inherent p-type conductivity and suitability for transparent electronics. Innovations in deposition techniques, such as aerosol-assisted chemical vapor deposition (AACVD), have enabled the fabrication of high-quality Cu₂O thin films [9]. NiO, known for its wide bandgap and chemical stability, has been explored as another viable p-type semiconductor. Liu et al. reported a remarkable 60-fold enhancement in hole mobility from 0.07 to 4.4 cm²/V·s primarily attributed to the high areal capacitance of the Al₂O₃ dielectric and the high-quality NiOx/Al₂O₃ interface, compared to conventional SiO₂-based dielectric [10].

[6] Chang, H., Huang, C.H., Matsuzaki, K., Nomura, K., Back-channel defect termination by sulfur for p-channel Cu2O thin-film transistors, ACS Appl. Mater. Interfaces. 2020, 12(46), 51581-51588.

[7] Hsu, S.M., He, J.C., Li, Y.S., Su, D.Y., Tsai, F.Y., Cheng, I.C., Effect of mechanical strain on electrical performance of flexible P-type SnO thin-film transistors, IEEE Trans. Electron Devices. 2019, 66(12), 5183-5186.

[8] Chae, M.G., Kim, J., Jang, H.W., Park, B.K., Chung, T.M., Kim, S.K., Han, J.H., High field-effect mobility and on/off current ratio of p-type ALD SnO thin-film transistor, ACS Appl. Electron. Mater. 2023, 5(4), 1992-1999.

[9] Ahmed, S., Shahid, M.M., Bakar, S.A., Arshed, N., Basirun, W.J., Fouad, H., Fabrication and characterization of SnO–Cu2O mixed metal oxide thin films for photoelectrochemical applications, J. Nanosci. Nanotechnol. 2020, 20(12), 7705-7709.

[10] Liu, A., Liu, G., Zhu, H., Shin, B., Fortunato, E., Martins, R., Shan, F., Hole mobility modulation of solution-processed nickel oxide thin-film transistor based on high-k dielectric. Appl. Phys. Lett. 2016, 108(23), 233506.

Reviewer 2 Report

Comments and Suggestions for Authors

The paper reports on the fabrication of SnO nanosheet pFET devices. The work is the continuation of the ongoing efforts by the group and demonstrates improvement in the pFET performance reported recently. The results are promising but incremental and the manuscript lacks in-depth discussion of the observed Physics. Below are some recommendations for improvement.

How many devices were fabricated and tested for each device without a passivation layer and without UV annealing, devices with a passivation layer but without UV annealing, and devices with both a passivation layer and UV annealing? (Fig.12)? Show the device statistics.

Fig.3(b) TEM cross-section does not show a sharp interface between subsequent layers. Was this TEM taken before or after annealing? We do not find in the discussion how the interface roughness affected the pFET performance b/c of the interface charges (e.g. trapping).

In Fig.6 (a) and (b) the field-effect mobility curves are different for 10 nm and 12 nm SnO. There is a shoulder around -0.75 V in Fig,6(b). Please comment on its origin.   

Fig.5, and 7 provide the legend for different I-V ch-tics. Does the Id increase monotonically with the VG?

Fig.8 explains the difference between the black curve ch-tic and green curve ch-tic.

In Fig.10(a) the field-effect mobility curves show the evolution of the shoulder at -0.75V, please comment on the observation.  

In Fig.11, were the test/reference and completed devised encapsulated?  If not, how will the encapsulation change the device's performance? How to mitigate the device “aging” process in the air when implementing the proposed pFET into the IC?

The use of the first principle quantum-mechanical calculation of SnO using Quatum-espresso is not clear. There is no discussion of the theoretical results/prediction nor comparison with the experimental results. Use it effectively or remove it from the paper.  

Major revisions are recommended.

Author Response

Thank you for the comments. To strengthen this paper, we have included your important comments in the revised manuscript (highlighted with yellow color).

  1. How many devices were fabricated and tested for each device without a passivation layer and without UV annealing, devices with a passivation layer but without UV annealing, and devices with both a passivation layer and UV annealing? (Fig.12)? Show the device statistics.

Ans: The device reproducibility can be found in our previous publications [34], [14], [18]. The μeff of 20.7 cm2/V·s is the highest value for SnO pFET with a typical standard deviation of 3.4 for 10 devices and ION/IOFF of 6.9 × 10⁶ with standard deviation of 87. Such device variability was also close to our reported works on SnON nFET [2].

[2] Pooja, P., Che, C. C., Zeng, S. H., Lee, Y. C., Yen, T. J., Chin, A., Outstanding high field‐effect mobility of 299 cm2 V−1s−1 by nitrogen‐doped SnO2 nanosheet thin‐film transistor, Adv. Mater. Technol. 2023, 8(7), 2201521.

[14] Shih, C.W., Chin, A., Lu, C.F., Su, W.F., Remarkably high hole mobility metal-oxide thin-film transistors, Sci. Rep. 2018, 8(1), 889.

[18] Zeng, S.H., Pooja, P., Wu, J., Chin, A., Impact of UV annealing on the hole effective mobility in SnO pFET, Sci. Rep. 2024, 14(1), 26256.

[34] Yen, T.J., Chin, A., Gritsenko, Exceedingly high performance top-gate p-type SnO thin film transistor with a nanometer scale channel layer, Nanomaterials. 2021, 11, 92.

  1. 3(b) TEM cross-section does not show a sharp interface between subsequent layers. Was this TEM taken before or after annealing? We do not find in the discussion how the interface roughness affected the pFET performance b/c of the interface charges (e.g. trapping).

Ans: The TEM image in Figure 3(b) has been taken after anneal. Interface roughness can increase hole scattering, especially at high charge density or effective field [22]. Interface charges at both top and bottom SnO interfaces results in mobility degradation and poor subthreshold performance. To lower the interface charge scattering, SiO2 with 6 nm thickness is deposited in between the channel SnO and gate dielectric HfO2 to reduce the remote phonon scattering from high-κ gate dielectric. After deposition of HfO2 and SiO2, the device undergoes thermal anneal in an N₂ environment at 400°C for 1 hour before SnO channel deposition. Thermal anneal at 400°C in N₂ improves the electrical quality of HfO₂ and SiO₂ dielectrics, reduces interface trap states and enhance the mobility. To lower the top interface charge scattering, SiO2 passivation was also applied.

[22] Takagi, S.I., Toriumi, A., Iwase, M, Tango, H., On the universality of inversion layer mobility in Si MOSFET's: Part I-effects of substrate impurity concentration, IEEE Trans. Electron Devices. 1994, 41, 2357 – 2362.

  1. In Fig.6 (a) and (b) the field-effect mobility curves are different for 10 nm and 12 nm SnO. There is a shoulder around -0.75 V in Fig,6(b). Please comment on its origin.

Ans:  Thank you for your observation. The shoulder-like non-smooth µFE feature is also found in Figure 10(a).

As shown in Figure 10(a), the shoulder-like non-smooth µFE feature in 7-nm thick SnO channel is attributed to extra scattering from defects, which can be improved by passivation and UV anneal. The shoulder-like non-smooth µFE feature in Figures 6(a) and 6(b) is due to thicker SnO channel, where gate electric field cannot fully deplete the SnO channel and defects. This is confirmed by the higher ION/IOFF in thicker SnO channel pFETs.

  1. 5, and 7 provide the legend for different I-V ch-tics. Does the Id increase monotonically with the VG?

Ans: The |Id| curves do not increase monotonically with the VG, which is due to the µeff roll off with increasing Qh [22].

[22] Takagi, S.I., Toriumi, A., Iwase, M, Tango, H., On the universality of inversion layer mobility in Si MOSFET's: Part I-effects of substrate impurity concentration, IEEE Trans. Electron Devices. 1994, 41, 2357 – 2362.

  1. 8 explains the difference between the black curve ch-tic and green curve ch-tic.

Ans:  Thank you for your suggestion. The black and green curves in Figure 8 represent |Id| and |Ig| characteristics, respectively. However, we realize that the legend might have been too small to be easily noticed. We have updated the figure with enlarged and clearer legends to distinguish |Id| and |Ig| properly.

  1. In Fig.10(a) the field-effect mobility curves show the evolution of the shoulder at -0.75V, please comment on the observation.  

Ans: Thank you for your observation. We already explained in above comment 2.

  1. In Fig.11, were the test/reference and completed devised encapsulated?  If not, how will the encapsulation change the device's performance? How to mitigate the device “aging” process in the air when implementing the proposed pFET into the IC?

Ans: Thank you for raising this important point.

The device shown in Figure 11 (a) was not encapsulated to show the ageing effect. The unpassivated control device exhibits significant degradation after two months of air exposure, with μFE dropping from 5.8 to 1.5 cm²/V·s. In contrast, the passivated device only shows a minimal decline from 20.3 to 19.2 cm²/V·s highlighting the strong protective role of our passivation approach. For future practical integration into ICs, external encapsulation using SiN will be applied on top Cu and low-κ interconnect back-end-of-line (BOE) layers to improve long-term stability. However, the SiO2 passivation directly on SnO pFETs must still be applied to prevent any moisture-related degradation during BOE process.

  1. The use of the first principle quantum-mechanical calculation of SnO using Quantum-espresso is not clear. There is no discussion of the theoretical results/prediction nor comparison with the experimental results. Use it effectively or remove it from the paper. 

Ans: We add the related explanation in the revised text:

First principle quantum-mechanical calculation using Quantum-espresso has been used to study the band structure of SnO and calculate its effective mass of hole (mh*). SnO bandgap of 0.62 eV and 0.18 m0, effective mass of hole (mh*) at ? points are obtained.

Figure 1. Energy band structures of SnO. CBM and VBM represent conduction band minimum and valence band maximum respectively.

Table 1. Comparison table of p-type metal-oxide semiconductor

p-type

Eg

Hole effective mass (mh*)

SnO

0.62 eV (Indirect)

0.18 m0

Cu2O [14]

2.1 eV (Direct)

0.65 m0

NiO [21]

3.6 eV (Indirect)

0.45 m0

[14] Shih, C.W., Chin, A., Lu, C.F., Su, W.F., Remarkably high hole mobility metal-oxide thin-film transistors, Sci. Rep. 2018, 8(1), 889.

[21] Zhang, Y. Thermal oxidation fabrication of NiO film for optoelectronic devices, Appl. Surf. Sci. 2015, 344, 33-37.

Reviewer 3 Report

Comments and Suggestions for Authors

The authors in the manuscript with ID nanomaterials-3574273 investigated the electrical performance and material properties of three types of devices and reported record performances (on-current/off-current of 6.9×106) of the oxide-semiconductor p-type FET (pFET). The three types of devices investigated were: devices without passivation layer and without UV annealing, devices with passivation layer but without UV annealing, and devices with both passivation layer and UV annealing. They used novel SiO2 surface passivation and ultraviolet (UV) light annealing. The outstanding outcome consisted in obtaining for the first time an oxide-semiconductor which has transistor's hole µeff reaches 20% of single-crystal Si pFET at the 7 nm ultra-thin body thickness. The authors also obtained remarkably high µeff of ~70 cm2 /V·s and 20.7 cm2 /V·s at hole density (Qh) of 1 × 1011 cm-2 and 5 × 1012 cm-2 , respectively.

This excellent SnO pFET can be further integrated with the record high mobility SnON nFET  on the integrated circuit backend forming a low-DC power Complementary Metal-Oxide-Semiconductor logic, which is useful for monolithic three-dimensional (M3D) integrated circuits (ICs).

The manuscript is well-written, it has adequate and relevant references, and the outstanding outcome in the field of Electronic Engineering  are of interest to all researchers  in nano IC manufacture process.

As a result, I agree with the publication of this review in its present form.

Author Response

Thank you for the excellent comments.

Reviewer 4 Report

Comments and Suggestions for Authors

This paper describes fabrication and optimization of p-channel tin oxide (SnO) field-effect transistors (SnO pFETs) using SiO2/HfO2/SiO2 as a passivation layer. It is shown that 7 nm thick SnO layers result in the best performing pFETs. The idea appears to be original and interesting. The performance improvement of the reported SnO pFETs is substantial. As such, I would like to recommend the article to be published in “Nanomaterials”. However, before the possible acceptance of the manuscript, I would like the authors to consider the following points:

  1. The paper has many typographical and grammatical errors which must be corrected before possible publication. I recommend the authors have a very careful check of their paper before the re-submission of their manuscript. For example:

Line 39: “…oxide-semiconductor pFET ARE reported…”.

Line 51: delete “subsequently”.

Line 59/60: “The device was applied by rapid thermal annealing (RTA) at 200°C in an N₂ environment for 30 seconds.” This sentence does not make sense.

Line 97: the thickness sequence is wrong.

Fig.1. Energy band STRUCTURE of SnO.

Line 117: “Good transistor pinch-off ARE observed at positive Vg.

etc.

  1. What exactly is shown in Figure 2(d)? This should be explained in the text.
  2. It is not clear how they patterned their SnO films.
  3. The authors write: “The gate leakage is only 4 × 10-6 A/cm2 that is due to the high-κ gate dielectric.” Actually, the leakage is rather high. The reasons for such high gate leakage should be discussed in the text and compared with other high-k materials of a comparable thickness.
  4. They should state the capacitance per unit area of their dielectric in nF/cm2 and explain how they calculated the field-effect mobility (and other device parameters) of their FETs.
  5. Fig. 5. Why the drain current is increasing in the saturation regime? Is it due to the high gate leakage of their devices? How the gate leakage can be reduced well below 10-6 A/cm2? This should be explained in the text.
  6. To support their conclusions, they shall discuss a possible influence of interface traps on the performance of their devices. How to calculate the interfacial charge trap density (DIT) the authors can see, for example, in IEEE Electron Device Letters, vol. 39, no. 3, pp. 375-378, March 2018, doi: 10.1109/LED.2018.2798061.
  7. It is not obvious how many transistors they fabricated, what was the device fabrication yield, how many FETs were tested, etc. Are the presented results from the best or representative devices?

In summary, there are several problems with the manuscript that must be addressed before possible publication.

Comments on the Quality of English Language

The paper has many typographical and grammatical errors which must be corrected before possible publication.

Author Response

Thank you for the comments. To strengthen this paper, we have included your important comments in the revised manuscript (highlighted with yellow color).

  1. The paper has many typographical and grammatical errors which must be corrected before possible publication. I recommend the authors have a very careful check of their paper before the re-submission of their manuscript. For example:

Line 39: “…oxide-semiconductor pFET ARE reported…”.

Line 51: delete “subsequently”.

Line 59/60: “The device was applied by rapid thermal annealing (RTA) at 200°C in an N₂ environment for 30 seconds.” This sentence does not make sense.

Line 97: the thickness sequence is wrong.

Fig.1. Energy band STRUCTURE of SnO.

Line 117: “Good transistor pinch-off ARE observed at positive Vg.

etc.

Ans: As suggested, the following changes have been made in the revised manuscript.

  1. What exactly is shown in Figure 2(d)? This should be explained in the text.

Ans: As suggested the following sentences has been added in the revised manuscript.

In Figure 2(d), the SiOₓ transition layer forms at the interface between SiO₂ and SnO. This SiOₓ originates from the low-temperature-deposited SiO₂ side due to oxygen diffusion or partial reduction of SiO₂ during processing. The interface contains various charges: oxide trapped charges deep in SiO₂, fixed oxide charges near the interface, and interface trapped charges at the SiOₓ/SnO boundary. These charges can affect device behavior by shifting threshold voltage and degrading carrier mobility.

  1. It is not clear how they patterned their SnO films.

Ans: A patterned mask was used for SnO films.  

  1. The authors write: “The gate leakage is only 4 × 10-6 A/cm2 that is due to the high-κ gate dielectric.” Actually, the leakage is rather high. The reasons for such high gate leakage should be discussed in the text and compared with other high-k materials of a comparable thickness.

Ans: As suggested, the reason of leakage and comparison of gate leakage of high-κ materials with comparable thickness has been added in the revised manuscript.

The gate leakage current of metal-gate/high-κ/SnO is significantly higher than standard metal-gate/high-κ/single-crystal Si Complementary Metal-Oxide-Semiconductor (CMOS) FET [23]. This is because the former is made at limited 400oC for back-end-of-line (BOE) process while the latter is fabricated at 1000oC. The limited thermal budget causes defects and trap-assisted leakage in high-κ gate dielectric. To further decrease the gate leakage current processed at low temperature, Atomic Layer Deposition (ALD) is useful but it takes long process time.

[23] Huang, C.H., Yu, D.S., Chin, A., Wu, C.H., Chen, W.J., Zhu, C.X., Li, M.F., Cho, B.J., Kwong, D.L., Fully silicided NiSi and germanided NiGe dual gates on SiO2/Si and Al2O3/Ge-on-insulator MOSFETs, IEEE IEDM Tech. Dig. 2003, 319-322.

  1. They should state the capacitance per unit area of their dielectric in nF/cm2 and explain how they calculated the field-effect mobility (and other device parameters) of their FETs.

Ans: As suggested, the capacitance per unit area of the dielectric has been changed to nF/cm2 and the method to calculate field effect mobility (μFE) in the revised manuscript.

Figure 4. C-V at 1 kHz and J-V characteristics of the Ni/6 nm-SiO2/45 nm-HfO2/TaN capacitor.

                        (1)

Where L and W are the channel length and width, Cox is the gate capacitance per unit area and gm is the transconductance.

  1. Fig. 5. Why the drain current is increasing in the saturation regime? Is it due to the high gate leakage of their devices? How the gate leakage can be reduced well below 10-6 A/cm2? This should be explained in the text.

Ans: In Figure 5, the drain current is increasing in the saturation regime. The drain current increasing trend is not due to gate leakage current, which is much lower than drain current (Figure 8). This is because the thicker SnO channel layer that cannot be fully pinched off by gate electric field, which is supported by the high IOFF.

To further decrease the gate leakage current processed at low temperature, Atomic Layer Deposition (ALD) is useful but it takes long process time.

  1. To support their conclusions, they shall discuss a possible influence of interface traps on the performance of their devices. How to calculate the interfacial charge trap density (DIT) the authors can see, for example, in IEEE Electron Device Letters, vol. 39, no. 3, pp. 375-378, March 2018, doi: 10.1109/LED.2018.2798061.

Ans: The interfacial charge trap density (Dit) [28], [29] has been calculated and it has been found that passivated SnO with UV anneal has a Dit of 4.6×1012 cm-2 eV-1, close to passivated SnO (4.7×1012 cm-2 eV-1) and better than SnO (8.2×1012 cm-2 eV-1).

[28] Cai, W., Park, S., Zhang, J., Wilson, J., Li, Y., Xin, Q., Majewski, L., Song, A., One-volt IGZO thin-film transistors with ultra-thin, solution-processed AlxOy gate dielectric, IEEE Electron Device Lett. 2018, 39(3), pp.375-378.

[29] Chang, M.F., Lee, P.T., McAlister, S.P., Chin, A., Low subthreshold swing HfLaO/pentacene organic thin-film transistors. IEEE Electron Device Lett. 2008, 29(3), 215-217.

  1. It is not obvious how many transistors they fabricated, what was the device fabrication yield, how many FETs were tested, etc. Are the presented results from the best or representative devices?

Ans: The μeff of 20.7 cm2/V·s is the highest value for SnO pFET with a typical standard deviation of 3.4 for 10 devices and ION/IOFF of 6.9 × 10⁶ with standard deviation of 87. Such device variability was also close to our reported works on SnON nFET [2]. The device reproducibility can be found in our previous publications [34], [14], [18].

[2] Pooja, P., Che, C. C., Zeng, S. H., Lee, Y. C., Yen, T. J., Chin, A., Outstanding high field‐effect mobility of 299 cm2 V−1s−1 by nitrogen‐doped SnO2 nanosheet thin‐film transistor, Adv. Mater. Technol. 2023, 8(7), 2201521.

[14] Shih, C.W., Chin, A., Lu, C.F., Su, W.F., Remarkably high hole mobility metal-oxide thin-film transistors, Sci. Rep. 2018, 8(1), 889.

[18] Zeng, S.H., Pooja, P., Wu, J., Chin, A., Impact of UV annealing on the hole effective mobility in SnO pFET, Sci. Rep. 2024, 14(1), 26256.

[34] Yen, T.J., Chin, A., Gritsenko, Exceedingly high performance top-gate p-type SnO thin film transistor with a nanometer scale channel layer, Nanomaterials. 2021, 11, 92.

Round 2

Reviewer 2 Report

Comments and Suggestions for Authors

This is a revised manuscript. The authors addressed the reviewer’s concerns/questions, Thank you! We recommend publishing the paper as is now.

Author Response

Thank you for the excellent comments.

Reviewer 4 Report

Comments and Suggestions for Authors

The authors addressed most of my concerns and efficiently dealt with the other major weaknesses of the previous version of the manuscript. As a result, I would like to recommend the paper for the publication in Nanomaterials.

Comments on the Quality of English Language

There are still several typographical and grammatical errors. I recommend (and expect) the authors to closely work with the editor and the typist to correct the mistakes before the final publication.

Author Response

Thank you for the excellent comments. The English in the revised manuscript is improved to express the research clearly.